# Quantitative FRET-FLIM-BlaM to Assess the Extent of HIV-1 Fusion in Live Cells

**DOI:** 10.3390/v12020206

**Published:** 2020-02-12

**Authors:** Irene Carlon-Andres, Sergi Padilla-Parra

**Affiliations:** Division of Structural Biology, University of Oxford, Wellcome Centre for Human Genetics, Headington, Oxford OX3 7BN, UK; irene@well.ox.ac.uk

**Keywords:** HIV-1 fusion, FRET, FLIM, live cell imaging, HIV infection, single cell analysis

## Abstract

The first steps of human immunodeficiency virus (HIV) infection go through the engagement of HIV envelope (Env) with CD4 and coreceptors (CXCR4 or CCR5) to mediate viral membrane fusion between the virus and the host. New approaches are still needed to better define both the molecular mechanistic underpinnings of this process but also the point of fusion and its kinetics. Here, we have developed a new method able to detect and quantify HIV-1 fusion in single live cells. We present a new approach that employs fluorescence lifetime imaging microscopy (FLIM) to detect Förster resonance energy transfer (FRET) when using the β-lactamase (BlaM) assay. This novel approach allows comparing different populations of single cells regardless the concentration of CCF2-AM FRET reporter in each cell, and more importantly, is able to determine the relative amount of viruses internalized per cell. We have applied this approach in both reporter TZM-bl cells and primary T cell lymphocytes.

## 1. Introduction

The human immunodeficiency virus type 1 (HIV-1) is an enveloped virus that fuses with target cells and releases the genome-containing capsid in the cytosol. Interaction and fusion with the host cell is mediated by the viral envelope glycoprotein (Env) [1]. This process is quite complex and ineffective as the majority of virions fail to complete fusion. Several approaches to detect and measure HIV-1 fusion have been developed during the last decades, and they comprise population-based assays (bulk assays) [2] and also more precise approaches such as real time single virus tracking (SVT) [3,4]. Both approaches have their strengths and weaknesses; for example, bulk assays average out the fusogenic behavior of viruses for a whole population of cells. This means that single cell information is not available. Other assays based on the unmixing of a lipophilic fluorophore embedded in the virus envelope [5] and different labeling strategies for SVT [4,6,7] have been developed to pinpoint the sites of virus fusion and its kinetics. These methods provide finer level of detail about single-virus temporal and spatial dynamics, but also require a lot of expertise in both virus labeling and image analysis and are not straightforward. The β-lactamase (BlaM) assay [8] represents a gold standard technique for measuring viral fusion in a population of cells. Nevertheless, the cell-based readout inherent to the BlaM assay presents a number of caveats such as the difficulty to determine the absolute number of successful fusion events and the need of fusion inhibitors to recover time-resolved data on virus fusion. Here, we present a new approach based on lifetime imaging to measure FRET able to detect with single cell accuracy HIV-1 fusion independently of the concentration of CCF2-AM reporter in each cell. This provides more reliable and precise BlaM assay quantification, compared to intensity-based FRET analysis. Moreover, both intensity and lifetime-based analysis show a linear dependence between the amount of exposed HIV-1 particles to cells and the shift in CCF2-AM fluorescence, providing information about the relative amount of fused HIV-1 particles per cell. This method complements existing ones able to measure a cell population HIV-1 fusogenicity or the point of fusion with single virus precision. 

## 2. Materials and Methods 

### 2.1. Cell Culture

TZM-bl cells (kind gift by Quentin Sattentau, University of Oxford) and Lenti-X 293T cells (Takara Bio, Clontech, Saint Germain en Laye, France) were grown using complete Dulbecco’s Modified Eagle Medium (DMEM) and DMEM F-12 (Thermo Fisher Waltham, MA, USA), respectively, both of which supplemented with 10% fetal bovine serum (FBS), 1% penicillin-streptomycin, and 1% L-glutamine. Cells were maintained in a 37 °C incubator supplied with 5% CO_2_. CD4+ T cell lymphoblasts were purified from human peripheral blood CD4+ T cells isolated from healthy donors [9]. Briefly, CD4+ T cells were isolated by negative selection (Ro-setteSep Human CD4+ T cell Enrichment Kit, Stemcell technologies, Vancouver, Canada) following the manufacturer’s procedure. Rested CD4+ T cells were kept overnight in complete medium (RPMI 1640 medium, Sigma-Aldrich, St. Louis, MO, USA; supplemented with 10% heat-inactivated fetal bovine serum; 50 U/mL of penicillin-streptomycin; 2 mM L-glutamine; 10 mM HEPES; 1 mM sodium pyruvate and 100 µM nonessential amino acids) prior to experimental use. 

### 2.2. Plasmids

The pR8ΔEnv plasmid, pcRev and Vpr-BlaM [8] expressing pMM310 vector were kindly provided by Greg Melikyan (Emory University). The plasmids encoding the JR-FL or the HXB2 envelope [10] glycoproteins were kindly provided by James Binley (Torrey Pines Institute for Molecular Studies). 

### 2.3. Virus Production

Pseudotyped HIV-1 particles were produced by transfecting Lenti-X 293T cells plated at ∼60%–70% confluency in T175 flasks. Cells were transfected using GeneJuice (Novagen, Waltford, UK) according to the manufacturer’s instructions. The transfection mixture was prepared using serum-free medium, Opti-MEM (Thermo Fisher, Waltham, MA, USA), and included the following: 2 μg of pR8ΔEnv (encoding the HIV-1 genome, harboring a deletion within Env), 2 μg of Vpr-BlaM expressing pMM310 vector, 1 μg of pcRev and 3 μg of the appropriate viral envelope (either the CCR5-tropic HIV-1 strain JR-FL or the CXCR4-tropic HXB2) or pcDNA3.1(+) vector (Invitrogen, Waltham, MA, USA) for No Env pseudovirus production. Transfection mixtures were then added to cells growing in complete DMEM F-12. At 16 h post-transfection, the medium was replaced with fresh complete DMEM F-12. At 72 h post-transfection, viral supernatants were removed from cells and filtered through a 0.45 μm filter (Sartorius Stedim Biotech, Royston, UK) before being concentrated using Lenti-X Concentrator (Takara Bio, Clontech, Saint Germain en Laye, France). Concentrated viral pellets were then resuspended in phenol red-free medium, FluoroBrite DMEM (Thermo Fisher, Waltham, MA, USA), aliquoted and stored at −80 ºC. 

### 2.4. Viral Titer

To titer the produced pseudoviruses, we used TZM-bl cells, which are HeLa-derived cells ectopically expressing CD4 and CCR5 receptors required for HIV-1 entry. These cells also contain a β-galactosidase gene under the control of an HIV-1 LTR. As a result of HIV-1 transcription and expression of HIV-1 Tat transactivation protein, infected cells expressing β-galactosidase are able to cleave the 5-bromo-4-chloro-3-indolyl-β-D-galactopyranoside (X-gal) substrate showing an easily identifiable blue appearance. Hence, 2 × 10^4^ TZM-bl cells were seeded in complete DMEM, in triplicate, in a flat-bottom 96-well plate and allowed to grow at 37 °C, 5% CO_2_. The day after, the medium was replaced with 10-fold serial dilutions of the produced pseudovirus particles. At 48h after viral addition, cells were then washed with PBS 1X and fixed with 4% paraform-aldehyde for 10 min. After an additional PBS 1X wash post-fixation, an X-gal (i.e., BCIG, or 5-bromo-4-chloro-3-indolyl-β-D-galactopyranoside) solution comprised of 500 mM K_3_(Fe(CN)_6_), 250 mM K_4_(Fe(CN)_6_), 1 M MgCl_2_, PBS 1X and 50 mg/mL Xgal was added to the cells and incubated for two hours at 37 °C in the dark. Afterwards, cells were washed with PBS 1X and subsequently imaged using transmitted white light and a 10X objective from a Leica DMi8 microscope (Leica Microsystems, Mannheim, Germany). The number of infection-positive cells (i.e., blue cells) was counted in triplicate and the number of infectious particles calculated per mL of viral preparation. 

### 2.5. BlaM Assay

At 24 h prior to the assay, TZM-bl cells were seeded at 2 × 10^4^ cells/well in µ-Slide 8-well (Cat.No:80826, Ibidi, Gräfelfing, Germany). On the day of assay, cells were washed with cold PBS 1X and viruses were added at the indicated multiplicity of infection (MOI) diluted in Fluorobrite DMEM supplemented with 2% FBS, in a final volume of 100 μL. Immediately following addition of virus harboring Vpr-BlaM, cells were placed at 4 °C for 1 h. Viral inoculum was then removed and cells were washed with PBS 1X to remove unbound viruses before replacing the medium with 100 μL of complete DMEM. In case of primary CD4+ T lymphocytes, cells were placed in 1.5 mL Eppendorf tubes and viruses were added onto cells in a final volume of 300 μL of cold Fluorobrite DMEM supplemented with 2% FBS. Cells were then placed at 4 °C for 30 min before spinning them at 4 °C, 600 g for 15 min. Excess of unbound viruses was removed and the medium was replaced with complete RPMI 1640 medium. Cells were then incubated for 90 min at 37 °C, 5% CO_2_ to allow viral entry. After 90 min, cells were loaded with CCF2-AM from the LiveBLAzer FRET B/G Loading Kit (Thermo Fisher, Waltham, MA, USA) and incubated at room temperature in the dark for 2 h. Finally, cells were washed with PBS 1X and fixed with 4% PFA for 15 min prior to imaging.

### 2.6. BlaM Assay Spectral and Fluorescence Lifetime Image Acquisition

TZM-bl cells loaded with CCF2-AM were excited using a 440 nm pulsed laser tuned at 80 MHz (PicoQuant, Berlin, Germany) coupled with single photon counting electronics (TCSPC). The emission spectra between 450–480 nm (blue emission, cleaved CCF2-AM) and 500–540 nm (green, uncleaved CCF2-AM) was detected by hybrid detectors pixel by pixel (512 × 512 frame size) using a Leica SP8 X-SMD microscope with the FALCON module, from Leica Microsystems (Mannheim, Germany). Areas of interest were chosen under a 63×/1.4 NA water objective. Frame acquisitions lasted ∼3 min to accumulate enough photons in order to perform double exponential fits.

### 2.7. BlaM Assay Spectral and Fluorescence Lifetime Image Analysis

To rule out artifacts due to photo-bleaching and insufficient signal to noise, only cells with at least 100–1000 photons per pixel and negligible amount of bleaching were included in the analysis after a 3 × 3 image binning. The acquired fluorescence decay of each pixel was deconvoluted with the instrument response function (IRF) and fitted with two-exponential theoretical models using Leica Application Suite X (LAS X) from Leica Microsystems. The fraction of interacting donor (f_D_) was calculated using Leica Application Suite X (LAS X) from Leica Microsystems and ImageJ software (https://imagej.nih.gov/ij/) following Equations (2) and (3) (below). The ratio of average intenstity was then calculated pixel by pixel using ImageJ software. Regions of interest corresponding to individual cells were selected using a semi-automated macro using Image J software and applied to calculate the average intensity ratio and fraction of interacting donor per cell. Individual values per cell were plotted after normalization to the threshold given by our negative control (No Env virions packaging Vpr-BlaM). 

One can define the catalytic reaction of the Vpr-BlaM cleaving the CCF2-AM substrate right after HIV-1 fusion as:(1)$$\left[E\right]+\left[S\right]\;\overset{K-1/\;K1}{↔}\;\left[ES\right]\overset{kcat}{\to }\left[E\right]+\left[P\right]$$
where *E* is the enzyme, in this case the Vpr-BlaM, *S* the substrate, in this case the CCF2-AM, and *P* the products (cleaved CCF2-AM, hydroxycoumarin + fluorescein, Figure 1). The Michaelis–Menten constant, *K_M_*, is defined as:(2)$$K_{M}=\;\frac{K_{-1\;}+\;K_{cat}}{K_{1}}=\;\frac{\left[E\right]\left[S\right]}{\left[ES\right]}$$

It has been shown [11] that there is a linear dependence between *K_M_* and [*E*] which linearly scales with [*E*] with a –1 slope in live cells. This experimental relationship led us to propose a linear model for this catalytic reaction that we could recapitulate in Figure 2. Importantly, Zotter and coleagues [11] showed that the slower diffusion coeffcient of the substrate (CCF2-AM) in the cytosol of live cells as compared to in-solution is the explanantion for this linear behavior. Both hydroxycourmarin and fluorescein should have similar diffusion coeficients regardless of the cell line utilized. In [11] it was also shown that the metabolic state of HeLa cells did not perturb this linear dependency nor the BlaM efficiency. 

In FLIM, for a single fluorophore in a homogeneous environment the fluorescecnce decay can be defined in the following way:(3)$$i\left(t\right)=k_{r}\left[S^{*}\right]=\;k_{r}{\left[S^{*}\right]}_{0}\exp\left\{\frac{-t}{\tau }\right\}$$
where the [*S*^*^] corresponds to hydroxycoumarin (the donor in the CCF2-AM complex, with fluorescein being the acceptor) in the excited state (right after excitation with the pulsed 440 nm laser there is a promotion of one photon from the hydroxycoumarin higher occupied molecular orbital (HOMO) to the lowest unoccupied hydroxycoumarin molecular orbital (LUMO)). *k_r_* is the kinetic constant of photon relaxation, which, in turn, emits light between the LUMO and the HOMO [12]. Finally, τ is the lifetime (in ns) for the photons residing in the LUMO.

According to the BlaM catalytic equation (Equation (1)), one can define a proportion of substrate (CCF2-AM) engaged in FRET (*f_D_*) and a proportion that is not (1 – *f_D_*) and define the next equation: (4)$$i\left(t\right)=\left(1-f_{D}\right)\exp\left\{-\frac{t}{\tau_{D}}\right\}+\;f_{D}\exp\left\{-\frac{t}{\tau_{F}}\right\}-Bkr$$

For Equations (3) and (4), τ is the lifetime, τ*_D_* the CCF2-AM donor lifetime, τ*_F_* the FRET lifetime and *f_D_* the fraction of CCF2-AM donor engaged in FRET right before cleavage (Figure 1). Time domain FLIM allows the simultaneous calculation of the fluorescence lifetime of the donor, the FRET lifetime, which implies *E* and *f_D_*; which in this case is the key parameter as *f_D_* is proportional to [*ES*]_0_ or uncleaved CCF2-AM. We therefore employed Equation (4) as a model to analyze all FLIM CCF2-AM images and recover *f_D_* pixel by pixel. The averaged *f_D_* per cell was then recovered and plotted against the MOI to recover the linear dependency for the CCF2-AM BlaM catalytic reaction. 

### 2.8. Statistical Analysis

All statistical analyses were performed using GraphPad Prism software version 8.3.0. Comparative analysis of pseudotyped HIV-1 fusion in TZM-bl or primary CD4+ T cells was performed by a two-way ANOVA and Sidak’s multiple comparisons test. To calculate the number of internalized viruses per cell, the mean values per condition were fitted to a simple linear regression curve. Frequency distribution histograms of the number of viral particles internalized per cell were calculated applying a bin width of 1.

## 3. Results

### 3.1. BlaM Calibration Curve

The BlaM assay is a commercially available system that relies on the ability of the β-lactamase enzyme to cleave a fluorescent substrate termed CCF2-AM, inducing a substantial shift in the emission spectrum. The substrate CCF2-AM consist of a FRET tandem, hydroxycoumarin (acting as donor) and fluorescein (acceptor), separated by a cephalosporin β-lactamase ring. In presence of β-lactamase, energy transfer from the donor to the acceptor by FRET is disrupted, resulting in a shift from green emission (peak at 520 nm) to blue emission (447 nm) (see schema in Figure 1). The BlaM assay can be used specifically to mesure HIV-1 fusion [8,13]. The CCF2-AM substrate can easily diffuse into the cytoplasm of a wide number of cell lines (i.e., TZM-bl) and primary T cells [8]. On the other hand, viral particles can be manipulated fusing the β-lactamase to the HIV-1 viral protein R (producing Vpr-BlaM), which is incorporated into nascent virions. Consequently, fusion of HIV-1 particles bearing Vpr-BlaM with CCF2-AM loaded cells results in the release of the β-lactamase enzyme in the cell cytoplasm, with subsequent cleavage and fluorescence modulation of the CCF2-AM substrate. 

The catalytic activity of the β-lactamase enzyme on the CCF2-AM substrate has been thoroughly studied in vitro and in live cells [11]. Zotter and coleagues showed that the catalytic efficiency of this particular reaction is inversely proportional to the β-lactamase concentration in vivo. Importantly, this relationship turned out to be linear. Hence, aiming to recover a calibration curve for BlaM applied to HIV-1 fusion that relates the amount of CCF2-AM cleaved with the relative number of internalized viral particles per cell, we designed an experiment in which we inoculated different amounts of HIV-1_JRFL_ particles harboring Vpr-BlaM (different MOIs) onto TZM-bl cells loaded with CCF2-AM substrate. As a negative control for HIV-1 fusion, we used viral particles lacking the envelope glycoprotein, unable to enter the cell (No Env) (Figure 2). Viruses were allowed to fuse with cells during 90 min after inoculum removal (see Section 2) and the catalytic reaction was allowed to progress for 2 h. Resulting fluorescence was recorded utilizing very sensitive single photon counting detectors, and the mean values of donor to acceptor intensity ratio (Figure 2A, left column) and the fraction of interacting donor (Figure 2A, right column) were calculated per cell. In case of intensity-based analyses, we considered cells to be fusion-positive when showing values above the mean plus 2 standard deviations (SD) of the No Env condition and below the mean minus 2 SD in the case of lifetime-based analysis. It is of note that cleavage of CCF2-AM by Vpr-BlaM results in higher values of intensity ratio due to increased donor emission intensity, and inversely, the catalytic activity of β-lactamase reduces the proportion of interacting donor. As expected, we observed higher numbers of fusion-positive cells when increasing the inoculated MOI onto cells (Figure 2B,C). When plotting the average of donor-to-acceptor intensity ratio versus inoculated MOI for each condition, we found a linear dependence, with R^2^ = 0.85 (Figure 2D). When utilizing lifetime imaging, we employed a double exponential method assuming that not all the CCF2-AM population was engaged in FRET, as described in Section 2 and reviewed by us [2]. When plotting the fraction of interacting donor (*f_D_*) [14] versus the inoculated MOI we found a linear dependence, with R^2^ = 0.87 (Figure 2E). Clearly, both intensity and lifetime-based analyses show a linear correlation between the amount of viral concentration in the inoculum (MOI) and the efficiency of CCF2-AM cleavage, as a readout of HIV-1 fusion. This linear dependency implies that it is possible to estimate the relative amount of fused HIV-1 particles per cell from CCF2-AM fluorescence data, providing more detailed information about the relative number of HIV-1 fusion events that occured in a single cell.

### 3.2. Single Cell FRET-FLIM-BlaM: Recovering the Relative Number of Fused Virus Per Cell in TZM-bl

Aiming to recover the relative amount of HIV-1_JRFL_ particles internalized per cell, we employed the above calibration curves for a second experiment using TZM-bl cells exposed to HIV-1_JRFL_ (MOI = 5). As before, we considered No Env HIV-1 particles as negative control (Figure 3). As expected, entry of HIV-1_JRFL_ particles induced higher values of intensity ratio due to increased donor emission intensity, compared to the negative control (Figure 3B, left chart), and inversely, we observed reduced proportion of interacting donor when exposing cells to HIV-1_JRFL_ particles (Figure 3B, right chart). In both types of analysis we could detect the same subpopulation of cells undergoing fusion (Figure 3C). Using the calibration curves obtained for intensity and lifetime FRET approaches, we could calculate the relative amount of fused HIV-1 particles per cell for both types of analysis (Figure 3D). Moreover, both approaches provided similar results in terms of relative number of fusion events in single cells, as shown in the micrographs in Figure 3D. Therefore, these results show that both intensity- and lifetime-based approaches, on the one hand, can be applied to measure HIV-1 fusion in the context of BlaM assays, and on the other hand, are capable of providing additional information about the relative level of successful fusion events in single cells. 

### 3.3. Single Cell FRET-FLIM-BlaM: Recovering the Relative Number of Fused Virus Per Cell in Primary T Cells

In order to test the validity of our quantitative FRET-FLIM-BlaM approach in a more physiological-relevant environment, we exposed HIV-1_HXB2_ particles to primary naïve T cells (Figure 4). As in previous experiments, we used No Env HIV-1 particles as negative control (Figure 4A,B). Exposure of HIV-1_HXB2_ particles to T cells induced higher values of intensity ratio and reduced the fraction of interacting donor compared to No Env particles (Figure 4B). As expected, only a small percentage of T cells were fusion positive (~5%), with similar results for both types of analyses (Figure 4C). We applied the calibration curves obtained for intensity and lifetime FRET approaches and calculated the relative amount of fused HIV-1_HXB2_ particles per cell, resulting in similar Gaussian distributions (Figure 4D). Overall, these results show that intensity- and lifetime-based analyses of the BlaM assay using microscopy have the potential to provide relevant information about the relative success of HIV-1 fusion in single TZM-bl cells and, more importantly, in primary T cells.

## 4. Conclusions 

Until now, the BlaM assay has been mostly utilized as a population-based assay with plate readers [3,13] and flow cytometers [15]. These are of course valid approaches but have a number of limitations. These apparatuses are often equipped with low-sensitivity analogic detectors (photo-multiplier tubes (PMT)), are used in large populations of cells and are unable to resolve the relative amount of fusogenic particles per cell. In the last few years, a lot of attention has been focused on quantitatively resolving protein expression and protein–protein interactions in single living cells [16]. These assays need to be coupled in virology with single cell resolution to evaluate the heterogeneity of infection in a cell population. New microscopy approaches are crucial to close the gap and put together different methodologies. In that respect, our group has been utilizing advanced imaging approaches in virology for the last decade and we are convinced that these new methodologies will be very useful to assess retrovirus fusion and infection with higher accuracy at the single cell level. To be able to quantify the relative number of internalized viral particles per cell in a highly physiological environment is the first step to couple these assays with other approaches, such as single cell RNAseq [16] or, more importantly, single molecule microscopy [17]. The methodology presented here provides more detailed information about HIV-1 fusogenicity and can be applied in a wide range of studies, i.e., single cell susceptibility to HIV-1 fusion in the context of metabolic studies, or in the presence/absence of specific cellular surface receptors to study their involvement in HIV-1 fusion. In this article, we present a new method for quantitative BlaM microscopy, which is more straightforward and quantitative for single cell fusion assays, population-based assays and drug screen tests in both reporter cells and primary cells. 

## Figures and Tables

**Figure 1 viruses-12-00206-f001:**
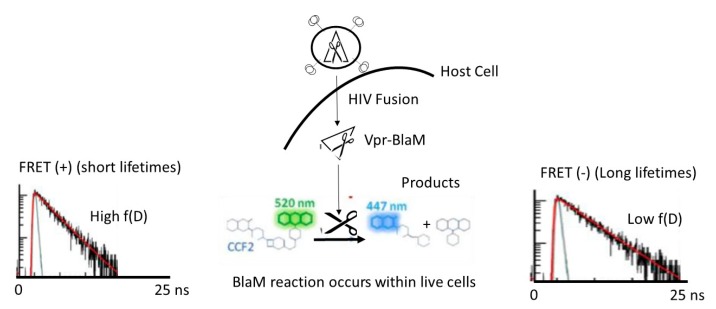
Schema of the β-lactamase (BlaM) assay applied to study human immunodeficiency virus type 1 (HIV-1) fusion utilizing intensity-based Förster resonance energy transfer (FRET) approaches and FRET fluorescence lifetime imaging microscopy (FLIM). HIV-1 pseudoparticles bearing the Vpr-BlaM are exposed to live cells expressing CD4 and co-receptors so that the HIV-1 particles are allowed to fuse with the host membrane and release their capsid into the host cytosol. The BlaM enzyme reaches the CCF2-AM FRET biosensor composed of hydroxycoumarin linked to fluorescein by the BlaM recognition domain. Once the BlaM enzyme starts the catalytic reaction, CCF2-AM is cleaved into hydroxycoumarin (donor) and fluorescein (acceptor) and FRET is disrupted. When employing FLIM, one can apply a two-exponential method that takes into account the situation in which not all fluorophore is engaged in FRET. A change from green to blue is also observed when looking at the visible emission spectra of CCF2-AM during the BlaM enzymatic reaction. The green emission (~520 nm) when exciting hydroxycoumarin at 405 nm comes from FRET; when CCF2-AM is cleaved hydroxycoumarin emits in the blue (~447 nm).

**Figure 2 viruses-12-00206-f002:**
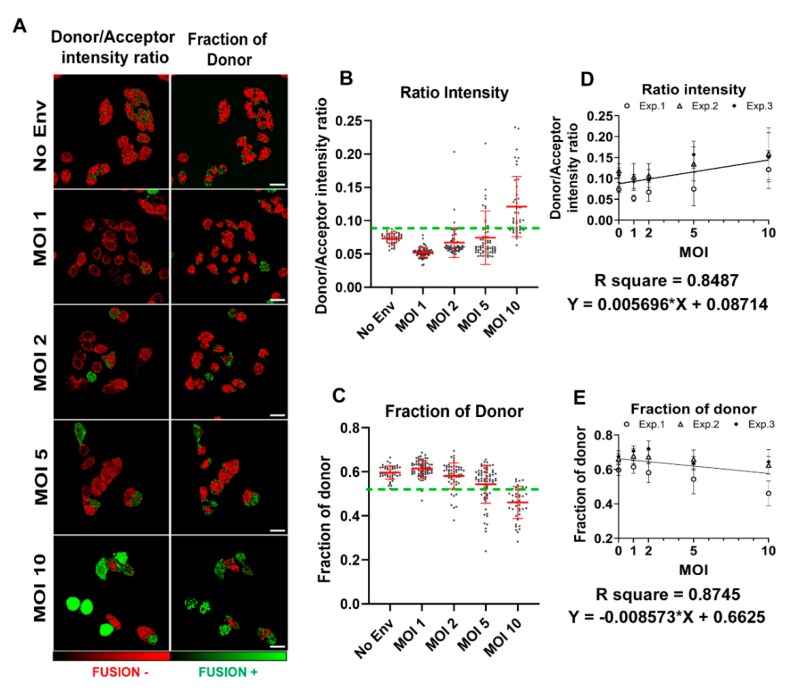
BlaM HIV-1 fusion calibration utilizing intensity-based FRET approaches and FRET-FLIM. (**A**) Images of the BlaM assay in TZM-bl cells exposed to different MOIs (1, 2, 5 and 10) of HIV-1_JFRL_ virions, analyzed using FRET intensity ratio and the fraction of interacting donor (*f_D_*). Images are pseudocolored in red (fusion negative) and green (fusion positive). Scale bars are 20 µm. The statistics coming from the two methods, (**B**) FRET intensity and (**C**) *f_D_*, are presented in which every dot represents a single cell (*n* > 50 cells per condition from one experiment). The green dotted line represents the threshold taken from TZM-bl cells exposed to HIV-1 without spikes (No Env) virions. (**D,E**) The calibration curves for both methods are presented together with the linear regression and equation/s. The calibration curves were obtained from the data points of three independent experiment (Exp. 1–3). Each symbol represents the mean and SD per condition, per experiment.

**Figure 3 viruses-12-00206-f003:**
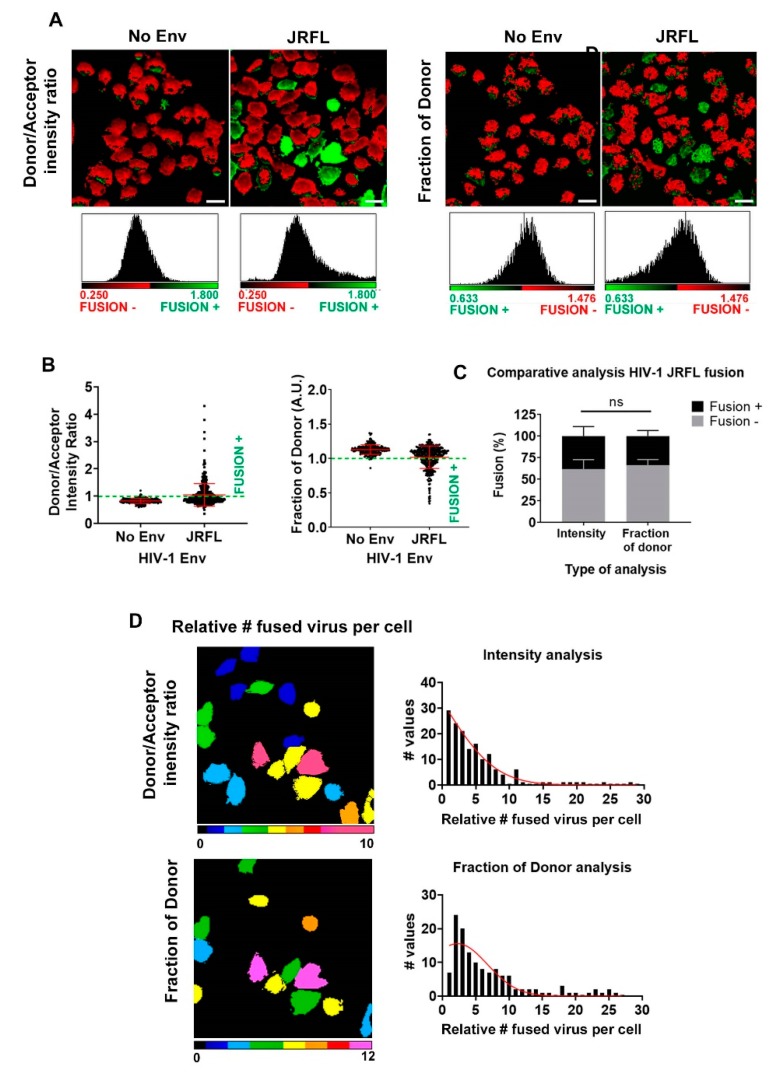
BlaM HIV-1 fusion quantification per cell in TZM-bl cells. (**A**) Micrographs for a population of TZM-bl cells exposed to HIV-1_JFRL_ virions (MOI = 5) and their corresponding pixel-by-pixel histograms are also presented for FRET-intensity-based and *f_D_* methods. The images are pseudocolored (red, fusion negative; green, fusion positive). In all cases, No Env HIV-1 virions were employed as a negative control and the CCF2-AM basal cleavage utilized as a reference for negative fusion. Scale bars are 20 µm. (**B**) The normalized statistics from three independent experiments for a population of cells with single cell accuracy are presented for the methods outlined above; *n* = 50–150 cells for each condition, for each experiment. Error bars are the standard deviation (SD) of the mean. (**C**) The overall percentage of fusion for a population of TZM-bl cells exposed to HIV-1_JFRL_ virions (MOI = 5) is presented for both methods. An average of ~25% was found in both cases. Error bars represent the SD of the mean from the three independent experiments. (**D**) Using calibration curves from Figure 2D,E, the relative amount of internalized HIV-1_JFRL_ virions per cell was obtained, again with single cell accuracy. Images are pseudocolored (cold colors represent relatively low amounts of internalized viral particles and warm colors, relatively high amounts of internalized particles). The frequency plots per cell are plotted for each approach (right column).

**Figure 4 viruses-12-00206-f004:**
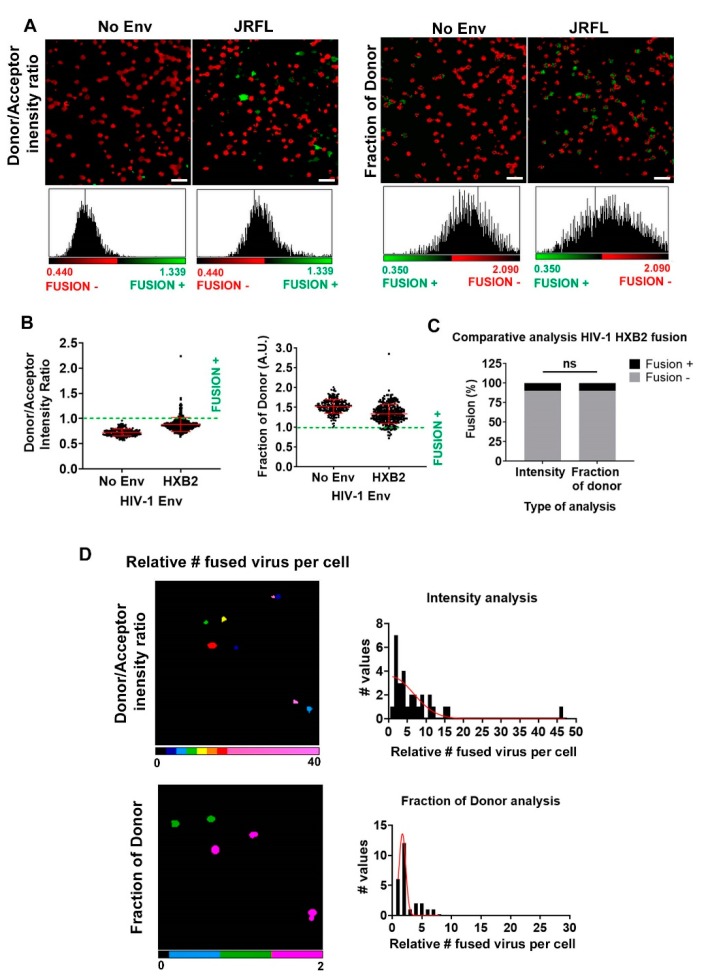
BlaM HIV-1 fusion quantification per cell in primary T cells. (**A**) Micrographs for a population of primary T cells exposed to HIV-1_HXB2_ virions and their corresponding pixel-by-pixel histograms are also presented for FRET-intensity-based and *f_D_* methods. In all cases, No Env HIV-1 virions were employed as a negative control and the CCF2-AM basal cleavage utilized as a reference for negative fusion. The images are pseudocolored (red, fusion negative; green, fusion positive). Scale Bars: 20 µm (**B**) The normalized statistics for a population of cells with single cell accuracy are presented for the methods outlined above; *n* > 240 cells for each condition. Results are from one experiment with cells from one donor. Error bars are the standard deviation (SD) of the mean. (**C**) The overall percentage of fusion for a population of TZM-bl cells exposed to HIV-1_HXB2_ virions is presented for both methods. An average of ~5% was found for intensity-based and *f_D_*. (**D**) Using each calibration curve, the relative amount of fused HIV-1_HXB2_ virions per T cell was obtained, again with single cell accuracy. Images are pseudocolored (cold colors represent relatively low amount of internalized viral particles and warm colors, relatively high amount of internalized particles). The frequency plots per cell are plotted for each approach (right column).

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
