# Peer review of "Quantitative FRET-FLIM-BlaM to Assess the Extent of HIV-1 Fusion in Live Cells"

_viruses, 2020, doi:10.3390/v12020206_

Round 1
Reviewer 1 Report
The manuscript by I. Carlon-Andres and S. Padilla-Parra addresses the problem of quantitative determination of HIV-induced fusion in cells. The study combines several experimental approaches, which results are critically compared. The authors conclude that the novel approach is more reliable and accurate.
I have several critical comments.
1) Page 2, line 65.
The last sentence of the paragraph is not complete.
2) Page 2, line 87, middle.
Probably, “with” should be deleted.
3) Page 7, Lines 107-127.
The physical aspect of the experiments should be clarified, probably, in a separate paragraph of the “Materials and methods” section. It would be useful to briefly describe what is CCF2, why it may be cleaved or uncleaved, what leads to the cleavage, how the CCF2 cleavage characterizes the infection process. Which substance was donor, which one was acceptor, what kinds of emission/absorption spectra they have, how they appear in the system in the course of the infection processes, why the donor lifetime correlated with MOI? Line 121: “fraction of
interacting donors” — interacting with what? Line 122: “The ratio of average intensity...” — the ratio of which value to which one? Line 126: “Individual values per cell were plotted...” — which value was plotted vs. which one?
4) Page 4, line 143: “plotting the average CCF2 ratio...”
The ratio of which value to which one?
5) Page 4, line 144: “we found a linear dependence with R2 = 0.79”
Why the data were fitted by the linear dependence? Two of five points (the left most, control and MOI = 1) do not intersect the line by their confidence intervals, meaning that the non-monotony of the dependence is statistically significant. Please, explain.
6) Page 4, line 150-151: “This approach, utilzing a double exponential fit turned out to be the most accurate as the average lifetime compared to the MOI (Figure 1, third row) gave R2=0.86.” The double exponential fit of which particular dependence was utilized? Why this approach is the most accurate? If the conclusion is made basing on comparison of R2
value, the validity of linear fit should be strictly proved. For example, one can imagine, that the “true” dependence should be quadratic with the minimum at MOI = 1 (by the way, this toy model correlates with non-monotonous dependence in the right upper plot of the Fig. 1.) In this case the comparison of quality of linear fits will lead to incorrect conclusion: the best linear fit corresponds to the bottom right plot, while the most close to the “true” quadratic dependence is the right upper plot. Please, comment and clarify.
Besides, the exact value of R2 should be clarified for this approach: R2 = 0.86 in the text, while R2 = 0.8866 ≈ 0.89 in the plot. There is a misprint in the word “utilzing”.
7) Page 5, Figure 1 and all other figures.
The readability would benefit if all panels of the figures were numbered (e.g. by letters a, b, ...) and each of the panels was explicitly described in the legends.
8) Page 6, line 184-191: “Indeed, the average MOI recovered when using the calibration curve for intensity FRET approaches was 2.92+/- 3.0 as compared with FRET-FLIM (fD) , 4.69+/-4.0. Clearly, the fD approach is closer to the MOI 5 utilized for this particluar experiment. It is interesting to see that only FRET-FLIM (fD) could resolve three different sub-distributions of MOI (fusogenicity) in single cells: one with a maximum MOI around 3, a second one with a maximum MOI around 7, and a third one with a very low frequency with MOI’s between 20 and
25 (Figure 2, second row, right panel). This third population is only apparent when utilizing the FRET-FLIM (fD) approach.”
The comparative analysis of the data of Fig. 2 should be clarified. The authors consider the third population with MOI’s between 20 and 25 (the second row, right panel) as statistically significant. On the same plot, the pike at MOI = 17 is substantially higher; its height is larger
than the height at MOI = 20-25 and larger than the relative height at MOI = 7 (the difference of the fD at MOI = 7 and at MOI = 6 or 8). If this pike (MOI = 17) is ignored for some reasons, than the pikes at MOI = 7 and MOI = 20-25 should also be ignored. Besides, there are relatively high pikes at MOI = 12-13 in the plots of the 1st and 3rd rows; there is no pike at the same MOI in the plot of the 2nd row. This may mean, that the approach based on fD analysis is incorrect, as it fails to resolve the pike observed by means of two alternative approaches. Please, comment and clarify.
9) Page 8, Figure 3.
The size of the labels of histograms (left column) should be increased at least 2 times — it is very difficult to read the numbers.
In the 2nd row, middle panel, the smallest value for control is about 1. However, on the histogram (2nd row, left panel) there are plenty of events with fD < 1. What were the reason and the criterion of this cutoff of data points?
10) Page 8, Line 218.
A typo in the word “gree”.
11) Page 9, Lines 228-230: “When comparing intensity-based FRET with the FRET-FLIM (fD) one could see that the error (dispersion SD) is bigger in the former (SD = 0.13) as compared to the latter (SD = 0.24)” The sentence seems to be self-contradicting.
12) Page 9, Line 267: “This research received no external funding from...” The sentence looks strange, as the authors declare no funding, but indicate a number of the Award (203141).
Author Response
The manuscript by I. Carlon-Andres and S. Padilla-Parra addresses the problem of quantitative determination of HIV-induced fusion in cells. The study combines several experimental approaches, which results are critically compared. The authors conclude that the novel approach is more reliable and accurate.
I have several critical comments.
1) Page 2, line 65.
The last sentence of the paragraph is not complete.
We thank the reviewer for this comment and have now included the missing information accordingly.
2) Page 2, line 87, middle.
Probably, “with” should be deleted.
We thank the reviewer for this comment and have now corrected the manuscript accordingly.
3) Page 7, Lines 107-127.
The physical aspect of the experiments should be clarified, probably, in a separate paragraph of the “Materials and methods” section. It would be useful to briefly describe what is CCF2, why it may be cleaved or uncleaved, what leads to the cleavage, how the CCF2 cleavage characterizes the infection process. Which substance was donor, which one was acceptor, what kinds of emission/absorption spectra they have, how they appear in the system in the course of the infection processes, why the donor lifetime correlated with MOI? Line 121: “fraction of interacting donors” — interacting with what? Line 122: “The ratio of average intensity...” — the ratio of which value to which one? Line 126: “Individual values per cell were plotted...” — which value was plotted vs. which one?
We thank the reviewer for pointing out the lack of information about CCF2-AM in the manuscript. We have now included a new figure (Figure 1) and a detailed description about the BlaM principle, the features of CCF2-AM and its application to the study of HIV-1 fusion in the results section (3.1). We have also explained more in detail the spectral and fluorescence lifetime image analysis in the material an methods section.
4) Page 4, line 143: “plotting the average CCF2 ratio...”
The ratio of which value to which one?
The average CCF2-AM ratio refers to the donor to acceptor intensity ratio. This clarification has been included in the text.
5) Page 4, line 144: “we found a linear dependence with R2 = 0.79”
Why the data were fitted by the linear dependence? Two of five points (the left most, control and MOI = 1) do not intersect the line by their confidence intervals, meaning that the non-monotony of the dependence is statistically significant. Please, explain.
This is an important point that should have been disussed in the original manuscript. The catalytic activity of the Beta-Lactamase (B-Lac) in live cells employing CCF2 as a substrate has been extensively studied [Zotter et al., 2017, JBC]. We now discuss the importance of the attenuated diffusion of the substrate (CCF2) in the cytosol of live cells. As demonstrated in Zotter et al., the catalytic efficiency of this particular enzymatic reaction k(cat)/K(M) decreases in live cells as compared to in vitro.
Importantly, the Michaelis Menten constant (k(M)) increases linearly with increased enzyme (B-Lac) concentration. This is the crux of our linear model as K(M) = [S][E]/[ES] and we can approximate [ES] to the fraction of interacting donor (f(D)). Indeed, our results show that there is a negative relationship between f(D) and the increased concentration of [E] (Figure 1) which in turn is proportional to the MOI.
In the case of the ratiometric approach this relationship also exists. Here, the linear dependency should also be proportional to the K(M), as the CCF2 ratio = [Acceptor]/[Donor].
In the case of the ratiometric approach, the linear behaviour is experimentally noiser as compared to the lifetime approach. This is true for many examples where FRET is evaluated in live cells [see Padilla-Parra and Tramier., 2012 Bioessays]. This error could be minimized when using spectral unmixing or the 3 cube filter approach (REF). Whichever the case, the ratiometric approach will always suffer from its dependency on the CCF2 concentration and the fact that some cells are more charged with CCF2 biosensor/substrate than others. This will always give an intrinsic heterogeneity that affects the error of the measurement. Another important aspect of ratiometric approaches is that when evaluating the CCF2 ratio; one obtains an apparent FRET efficiency that is a convolution of the true FRET efficiency and the fraction of interacting donor (f(D)) (Padilla-Parra et al., 2008 or Padilla-Parra and Tramier 2011). All these things affect the experimentally proven linear dependency between the catalytic efficiency and the B-Lac concentration (see figure 3C in Zotter et al., 2017).
We have now discussed these aspects in the text L186 – 195 and cited the literature.
6) Page 4, line 150-151: “This approach, utilzing a double exponential fit turned out to be the most accurate as the average lifetime compared to the MOI (Figure 1, third row) gave R2=0.86.” The double exponential fit of which particular dependence was utilized? Why this approach is the most accurate? If the conclusion is made basing on comparison of R2 value, the validity of linear fit should be strictly proved. For example, one can imagine, that the “true” dependence should be quadratic with the minimum at MOI = 1 (by the way, this toy model correlates with non-monotonous dependence in the right upper plot of the Fig. 1.) In this case the comparison of quality of linear fits will lead to incorrect conclusion: the best linear fit corresponds to the bottom right plot, while the most close to the “true” quadratic dependence is the right upper plot. Please, comment and clarify.
There has been a confusion between the model utilized to fit the fluorescence decay of the CCF2 coming from the FLIM experiments and the linear dependency of the catalytic efficiency with the B-Lac concentration (which is proportional to the increased MOI used; as explained above).
The double experimental approach to fit the fluorescence decay is utilized in this context. We consider a model where a two population model is considered (a subpopulation that intereacts [CCF2] and therefore gives a positive FRET; and a subpopulation that is not engaged in FRET (E + P)). When HIV-1 fusion occurs the Vpr-B-Lac will start the reaction to convert the substrate (CCD2) into products (E + P) and therefore the fraction of interacting donor (which corresponds the relative concentration of CCF2) will decrease.
In this scenario the average lifetime will increase and the corresponding subpopulation engaged in FRET (CCF2) will decrease.
We now define this model and introduce the equations for the double exponential approach to fit the fluorescence decay (FLIM data). L137 – L196
Besides, the exact value of R2 should be clarified for this approach: R2 = 0.86 in the text, while R2 = 0.8866 ≈ 0.89 in the plot. There is a misprint in the word “utilzing”.
We apologize for this error. We have now simplified the article and only show the model with two exponentials and the calculation of the fraction of interacting donor (CCF2, hydroxycoumarin engaged in FRET with fluorescein). We have further commented this point in response to reviewer 3.
7) Page 5, Figure 1 and all other figures.
The readability would benefit if all panels of the figures were numbered (e.g. by letters a, b, ...) and each of the panels was explicitly described in the legends.
We agree the figures suffer from clarity and have now numbered panels accordingly.
8) Page 6, line 184-191: “Indeed, the average MOI recovered when using the calibration curve for intensity FRET approaches was 2.92+/- 3.0 as compared with FRET-FLIM (fD) , 4.69+/-4.0. Clearly, the fD approach is closer to the MOI 5 utilized for this particluar experiment. It is interesting to see that only FRET-FLIM (fD) could resolve three different sub-distributions of MOI (fusogenicity) in single cells: one with a maximum MOI around 3, a second one with a maximum MOI around 7, and a third one with a very low frequency with MOI’s between 20 and 25 (Figure 2, second row, right panel). This third population is only apparent when utilizing the FRET-FLIM (fD) approach.”
The comparative analysis of the data of Fig. 2 should be clarified. The authors consider the third population with MOI’s between 20 and 25 (the second row, right panel) as statistically significant. On the same plot, the pike at MOI = 17 is substantially higher; its height is larger than the height at MOI = 20-25 and larger than the relative height at MOI = 7 (the difference of the fD at MOI = 7 and at MOI = 6 or 8). If this pike (MOI = 17) is ignored for some reasons, than the pikes at MOI = 7 and MOI = 20-25 should also be ignored. Besides, there are relatively high pikes at MOI = 12-13 in the plots of the 1st and 3rd rows; there is no pike at the same MOI in the plot of the 2nd row. This may mean, that the approach based on fD analysis is incorrect, as it fails to resolve the pike observed by means of two alternative approaches. Please, comment and clarify.
We thank the reviewer for this comment. It is indeed difficult to assess the correctness of one approach based on how many sub-populations of the number of internalized particles can be resolved. We have therefore toned down the description of the distribution/s for the ratiometric approach and the fD-FLIM approach describing how many maxima we could see in each distribution and fitting this maxima to a gaussian distribution.
9) Page 8, Figure 3.
The size of the labels of histograms (left column) should be increased at least 2 times — it is very difficult to read the numbers.
We apologize for the lack of clarity in the labels of histograms and have now modified the size of the graphs accordingly.
In the 2nd row, middle panel, the smallest value for control is about 1. However, on the histogram (2nd row, left panel) there are plenty of events with fD < 1. What were the reason and the criterion of this cutoff of data points?
We thank the reviewer for this comment and have now included a more detailed description of the criteria followed to determine the fusion threshold in the results section, figures 2C and 2D, in case of intensity-based analyses, we considered cells to be fusion-positive when showing values above the mean plus 2 standard deviations (SD) of the No Env condition, and below the mean minus 2 SD in the case of lifetime-based analysis.
10) Page 8, Line 218.
A typo in the word “gree”.
We thank the reviewer for this comment and have now corrected the manuscript accordingly.
11) Page 9, Lines 228-230: “When comparing intensity-based FRET with the FRET-FLIM (fD) one could see that the error (dispersion SD) is bigger in the former (SD = 0.13) as compared to the latter (SD = 0.24)” The sentence seems to be self-contradicting.
We thank the reviewer for this comment and have now clarified this point in the new version of the text
12) Page 9, Line 267: “This research received no external funding from...” The sentence looks strange, as the authors declare no funding, but indicate a number of the Award (203141).
We apologize for this error and have modified the sentence consequently.
Reviewer 2 Report
In this manuscript, the authors describe a novel modification of a standard HIV-1 cell entry assay, which is frequently used in the field, mainly for bulk measurements. The data indicate that the described approach may represent a more sensitive and more accurate method to determine relative HIV entry efficiency than the original BlaM assay. I do have concerns regarding the current form of the manuscript, however.
The abstract states that the assay presented does allow determining the absolute number of virus particles that have undergone cytosolic entry in a given cell (by relatively fast and easy one-well measurement). If this was the case, it would be a significant advancement, but I am not convinced that it is backed up by the data. Bona fide validation by an alternative approach to count fused particles per cell is not presented (for this reason, the term ‘validation’ should be changed to tested or applied). The authors simply take the MOI determined in an infectivity assay as a synonym for number of fused particles per cell and use it to calibrate the standard curves in Fig 1. This is not correct and represents sort of a circular definition. In fact, one main reason why one would want to quantitate the number of particles that have undergone cytosolic entry is to correlate this number to the efficiency of subsequent steps and eventually to the successful infection of a cell under certain conditions. As far as I understood it, the facts that there is a lower SD and better R2 value in the standard curves (see also point 2) and that the average is closer to the MOI for the TZM-bl cells are taken to conclude that the calculated number must be correct. I do not believe one can make this point.
The manuscript title emphasizes usefulness of the assay in primary T cells, it is not apparent to me from Figure 3 that this is the case. Please clearly explain how the reader can appreciate that FRET-FLIM is more precise in assigning fusion positive cells. Why is it just ‘interesting’ that the average lifetime analysis does not yield a useful result? How is it possible to conclude that 2.62 is more accurate than 6.08 when the ground truth is unknown in this case (the statement ‘is perhaps an overestimation’ is an opinion rather that a data-based description)? These points require clarification.
The manuscript lacks information regarding how many independent experiments were performed per condition (only one?) how fields of view were selected for analysis, which proportion of cells met the criteria from line 116-117, and how many cells were counted per sample. Without this information, soundness of the results is difficult to assess. Furthermore, in the data sets for MOIs 1-5 in Figure 1 only a small number of cells analyzed displays values beyond the threshold set to classify fusion positive cells; consequently, 3 out of 4 mean values used to generate most of the standard curves are within the range of fusion negative cells and the curves do not appear to cover a linear range. This does in my view prevent robust quantification.
Even though this is a short article the scope and selection of references is quite unusual. Nine out of the 13 papers cited either include the corresponding author or are from his own lab. This does not do justice to the field of of quantitative HIV entry analysis (imaging and biochemical), with contributions from many labs over the past two decades or so. Referencing throughout the manuscript needs to be significantly redone, more clearly crediting the work of others. Also, references in Materials and Methods are missing.
I cannot suggest an easy way to amend point 1. On the other hand, I believe a more sensitive and more robust single cell assay to determine relative HIV pseudotype fusogenicity compared to the standard ratiometric assay (at least in TZM-bl cells) may merit publication as a short report, provided that the statements regarding absolute numbers of particles are removed (and provided the other points are addressed). Throughout the manuscript, the authors are unclear in the distinction between bulk, single cell, or single particle analyses. The manuscript describes a single cell analysis and I suggest that it is presented and discussed as such, compared to other single cell analyses for HIV entry.
Minor points:
Accuracy and definitions: I believe that the clarity of the manuscript suffers significantly from inaccuracies in terminology. Terms and phrases that describe different things are used interchangeably, which I find rather confusing: a) HIV and virus: the title and abstract indicate that this is a manuscript on HIV-1, data and primary references exclusively address HIV. In contrast, the introduction refers to enveloped viruses in general, which results in inaccurate statements (e.g. ‘virus fusion is complex and ineffective’ without any details or references). I suggest to revise the introduction, to either be more specific regarding other viruses (which should then also be addressed in the discussion) or focus only on HIV to avoid confusion. b) MOI and fused particles per cell: needs to be addressed, see main point 1. c) single cell and single virus particle analysis: needs to be addressed throughout the manuscript, see also main point 1. d) the word precise is used when describing assays which are not per se more precise but yield different or more detailed information. FLIM analysis: at least representative raw data sets illustrating the FLIM analysis should be presented as a supplement. The authors refer to a review by them (ref 1). to explain the analysis, but since this is the key of the publication, some more detailed explanation and example data and fits are needed so that the reader can assess the information (e.g, I struggle to follow the description of results in lines 153-161 without such an illustration). Lines 38 ff: This is relevant for the rationale of the study, but I do not clearly understand what the authors state here. Which ‘prior knowledge’ is needed for ratiometric measurements, but not for the proposed method? What does ’might not represent the best approach’ mean specifically? Please clarify. Lines 62-65: the paragraph is not finished. Also, the source of the plasmids mentioned is not the lab that first described these constructs. Original sources need to be cited for each plasmid in order to clearly identify which plasmids were used. Line 90: ‘was counted in triplicate’: three independent experiments or triplicate counting of the same sample? Lines 187 ff and Fig. 2.: based on small irregularities in a fairly limited data set (see point 2) the authors propose three peaks of 3, 7 and 20-25 particles per cell. Even apart from major points 1 and 2 this is an overstatement, which I would ask to rephrase. Line 246: Flow cytometry is not a bulk method, but a single cell analysis Please rephrase. Line 256: what does ‘high resolution’ mean in this context? Please clarify.
There are a few typos (media instead of medium; subscript/superscript not executed, extra words, reference not found in the reference list). Please recheck manuscript for errors.
Author Response
Reviewer 2
In this manuscript, the authors describe a novel modification of a standard HIV-1 cell entry assay, which is frequently used in the field, mainly for bulk measurements. The data indicate that the described approach may represent a more sensitive and more accurate method to determine relative HIV entry efficiency than the original BlaM assay. I do have concerns regarding the current form of the manuscript, however.
The abstract states that the assay presented does allow determining the absolute number of virus particles that have undergone cytosolic entry in a given cell (by relatively fast and easy one-well measurement). If this was the case, it would be a significant advancement, but I am not convinced that it is backed up by the data. Bona fide validation by an alternative approach to count fused particles per cell is not presented (for this reason, the term ‘validation’ should be changed to tested or applied). The authors simply take the MOI determined in an infectivity assay as a synonym for number of fused particles per cell and use it to calibrate the standard curves in Fig 1. This is not correct and represents sort of a circular definition. In fact, one main reason why one would want to quantitate the number of particles that have undergone cytosolic entry is to correlate this number to the efficiency of subsequent steps and eventually to the successful infection of a cell under certain conditions.
We have now corrected the statement “the assay presented does allow determining the absolute number of virus particles that have undergone cytosolic entry in a given cell” in the introduction and argued that both intensity and lifetime-based analysis show a linear dependence between the inoculated viral MOI and the cleavage of CCF2-AM as a surrogate of HIV-1 fusion, which provides information about the relative amount of fused HIV-1 particles per cell.
As far as I understood it, the facts that there is a lower SD and better R2 value in the standard curves (see also point 2) and that the average is closer to the MOI for the TZM-bl cells are taken to conclude that the calculated number must be correct. I do not believe one can make this point.
We agree with the reviewer’s point of view because we have found a relationship between the catalytic activity of BlaM through CCF2 cleavage and the relative number of internalized particles. We cannot conclude that this relative number per cell could correspond to the absolute number of infectious particles per cell and have now toned down these claims in the manuscript.
The manuscript title emphasizes usefulness of the assay in primary T cells, it is not apparent to me from Figure 3 that this is the case. Please clearly explain how the reader can appreciate that FRET-FLIM is more precise in assigning fusion positive cells.
Why is it just ‘interesting’ that the average lifetime analysis does not yield a useful result? How is it possible to conclude that 2.62 is more accurate than 6.08 when the ground truth is unknown in this case (the statement ‘is perhaps an overestimation’ is an opinion rather that a data-based description)? These points require clarification.
We have now simplified the article and only show the model with two exponentials and the calculation of the fraction of interacting donor (CCF2, hydroxycoumarin engaged in FRET with fluorescein). We have further commented this point in response to reviewer 3.
The manuscript lacks information regarding how many independent experiments were performed per condition (only one?) how fields of view were selected for analysis, which proportion of cells met the criteria from line 116-117, and how many cells were counted per sample. Without this information, soundness of the results is difficult to assess. Furthermore, in the data sets for MOIs 1-5 in Figure 1 only a small number of cells analyzed displays values beyond the threshold set to classify fusion positive cells; consequently, 3 out of 4 mean values used to generate most of the standard curves are within the range of fusion negative cells and the curves do not appear to cover a linear range. This does in my view prevent robust quantification.
We apologize for the lack of information regarding number of cells analyzed and number of independent experiments that were performed, and have been now included this information in the figure legend of each figure. We would like to clarify that values used to generate standard curves included both, fusion negative and positive cells, as we observe a modulation of CCF2 fluorescence using increasing concentrations of viruses in both population of cells, and we consider that the criterion to stablish a threshold to define fusion negative vs positive cells relative to the No Env condition is subjective (i.e. the mean of the No Env plus two SD in the case of intensity-based analysis).
Even though this is a short article the scope and selection of references is quite unusual. Nine out of the 13 papers cited either include the corresponding author or are from his own lab. This does not do justice to the field of of quantitative HIV entry analysis (imaging and biochemical), with contributions from many labs over the past two decades or so. Referencing throughout the manuscript needs to be significantly redone, more clearly crediting the work of others. Also, references in Materials and Methods are missing.
We apologize for the extensive self-referencing. Most of the ideas developed in the text are based on previous reports and thought it could be informative to the reader. We have now included other original reports that were also crucial and apologize to have them omitted in the first version. We have included all references in Materials and Methods to guide the reader.
I cannot suggest an easy way to amend point 1. On the other hand, I believe a more sensitive and more robust single cell assay to determine relative HIV pseudotype fusogenicity compared to the standard ratiometric assay (at least in TZM-bl cells) may merit publication as a short report, provided that the statements regarding absolute numbers of particles are removed (and provided the other points are addressed). Throughout the manuscript, the authors are unclear in the distinction between bulk, single cell, or single particle analyses. The manuscript describes a single cell analysis and I suggest that it is presented and discussed as such, compared to other single cell analyses for HIV entry.
We thank the reviewer for this comment. Indeed, these concepts were unclear. In this new version throughout the manuscript we talk about the relative number of HIV particles internalized per cell. This value, even if it is relative was found to be related to the amount of CCF2 cleaved (as shown in the material and methods). Importantly, the single cell resolution of this relative value of particles internalized per cell is very informative and important. Especially when related to other parameters like the relative concentration of receptors and co-receptors or other restricition factors that might vary in single cells.
Minor points:
Accuracy and definitions: I believe that the clarity of the manuscript suffers significantly from inaccuracies in terminology. Terms and phrases that describe different things are used interchangeably, which I find rather confusing: a) HIV and virus: the title and abstract indicate that this is a manuscript on HIV-1, data and primary references exclusively address HIV. In contrast, the introduction refers to enveloped viruses in general, which results in inaccurate statements (e.g. ‘virus fusion is complex and ineffective’ without any details or references). I suggest to revise the introduction, to either be more specific regarding other viruses (which should then also be addressed in the discussion) or focus only on HIV to avoid confusion.
We thank the reviewer for this comment. We have now focused the introduction on HIV-1 to avoid confusion and discussed that the presented approach can potentially be applied to the study of fusion of other enveloped viruses.
b) MOI and fused particles per cell: needs to be addressed, see main point 1. c) single cell and single virus particle analysis: needs to be addressed throughout the manuscript, see also main point 1.
We have now addressed these points in response to main point 1.
d) the word precise is used when describing assays which are not per se more precise but yield different or more detailed information. FLIM analysis: at least representative raw data sets illustrating the FLIM analysis should be presented as a supplement. The authors refer to a review by them (ref 1). to explain the analysis, but since this is the key of the publication, some more detailed explanation and example data and fits are needed so that the reader can assess the information (e.g, I struggle to follow the description of results in lines 153-161 without such an illustration).
We apologize for the lack of clarity explaining the principle of BlaM assay and FLIM analysis. This lack of information has also been pointed by reviewer 1 (see reviewer 1, points 3 and 6) and we have now included the information accordingly.
Lines 38 ff: This is relevant for the rationale of the study, but I do not clearly understand what the authors state here. Which ‘prior knowledge’ is needed for ratiometric measurements, but not for the proposed method? What does ’might not represent the best approach’ mean specifically? Please clarify.
We thank the reviewer for this comment. We have modified the text in the introduction to avoid confusion.
Lines 62-65: the paragraph is not finished.
We thank the reviewer for this comment. We have modified the text in the introduction to avoid confusion.
Also, the source of the plasmids mentioned is not the lab that first described these constructs. Original sources need to be cited for each plasmid in order to clearly identify which plasmids were used.
We have now thanked the original labs that provided these plasmids (Binley and Melikyan). Several publications have been used with these plasmids and we have employed these resources in many publications.
Line 90: ‘was counted in triplicate’: three independent experiments or triplicate counting of the same sample?
They were three independent experiments
Lines 187 ff and Fig. 2.: based on small irregularities in a fairly limited data set (see point 2) the authors propose three peaks of 3, 7 and 20-25 particles per cell. Even apart from major points 1 and 2 this is an overstatement, which I would ask to rephrase.
This statement has also been pointed out by reviewer 1 (see point 8) and have now be amended.
Line 246: Flow cytometry is not a bulk method, but a single cell analysis Please rephrase. Line 256: what does ‘high resolution’ mean in this context? Please clarify.
We agree with this comment and have corrected the text accordingly
There are a few typos (media instead of medium; subscript/superscript not executed, extra words, reference not found in the reference list). Please recheck manuscript for errors.
We thank the reviewer for this comment and have now corrected the manuscript accordingly.
Reviewer 3 Report
This article presents a new adaptation of BlaM assay to quantify the efficacy of the HIV-1 virus entry into the host cells. The authors adapt the BalM technique based on the b-Lactamase trans incorporation (via its fusion to Vpr) into the HIV-1 virus during the production step. During the infection, when the HIV-1 virions successfully fuse their membrane with the plasma membrane of the host cell the BlaM is released to the cell cytoplasm where it cleaves the CCF2 molecules (composed of two fluorophores constituting a FRET pair). This cleavage is typically monitored by measuring a spectral shift of the CCF2 fluorescence emission or a ratio of the fluorescence intensity of the donor and acceptor. In this article the authors propose a more accurate readout based on the analysis of the fluorescence lifetime.
General questions:
In my opinion the article lacks a clear description of the main principle of the BalM method used for the monitoring of the HIV-1 fusion as well as of the experimental approaches. I suggest to include in the introduction or at the very beginning of the results section, a clear scheme of the cell infected by Vpr-BalM containing virus, the release of the BalM upon the viral fusion and the cleavage of CCR2. Also the authors don’t mention the FRET pair (hydroxycoumarin/fluorescein) of the CCR2. In the figure 2 and 3 the authors quantify the number of viral particles in the cells at 90 minutes +2h post infection using a calibration curves based on the MOI inferred from the expression of b-gal in TZM-bl cells 48 hours post infection (with 24 hours of virus/cell contact). This implies that all the viral particles enter the cell within the first hours of infection? Are the % of infected cells the same is the viruses are washed out after initial 90 minutes incubation? Could the authors explain better the relationship between the MOI measured at 48 hours post infection and the number of intracellular viral particles determined in their experiments? When two population analysis is used for the FLIM/FRET data, were all the fit parameters free or one of the lifetimes was fixed? The data analysis should be better described in the text. Part 3.2 and 3.3 lack a clear description of the experimental procedure and repeatability of the experiments. How many independent samples were prepared? How many cells were analyzed in each condition? In the part 3.3 the authors determine the number of viruses that successfully enters the T macrophages using a calibration curve that was done with TZM-bl cells? However depositing the same quantity of the virus on the cells the % of infection, and thus the calibration curve is different for the T Cells comparing to TZM-bl cells. In the data in figure 1, values presented for the ratiometric and fraction donor approach are lower in the first infected sample (MOI 1) than the for the “No envelope” control. This point should be commented.
Concerning the data in figure 3, it is not clear why does the donor lifetime decrease when the HIV-1 viruses enter into the T Cells? The fusion should be accompanied by an increase of the average donor lifetime due to the loss of the FRET. This point should be better commented since it compromises the validation of the method.
Minor points:
Material and Method Section: The source of the HXB2 envelope is missing Annotations of the inserts (A,B,C….) in figures 1 and 3 are missing. In the figure 1 I suggest to respect the order of the presentation of the three analysis approaches in the micrographs and in the statistics. i.e. the cell images are organized from the left to the right D/A intensity ratio, Donor lifetime, fraction of the donor, then the graphs should follow the same order from the top to the bottom. For the better understanding of the cell images in the figure 1 I suggest to add the colorcode as it is done in figures 2 and 3. The manner to define the threshold (green dotty line) for the discrimination of the fusion+ cells is not clear. Authors say that this level corresponds to cells exposed to No Env virions, but the values corresponding to these No Env virions are also present in the figures as a negative control in line 1. Does the threshold represent the max (or min for the f(D)) value measured for the No Env? This point should be better explained in the text.
Author Response
Reviewer 3
This article presents a new adaptation of BlaM assay to quantify the efficacy of the HIV-1 virus entry into the host cells. The authors adapt the BalM technique based on the b-Lactamase trans incorporation (via its fusion to Vpr) into the HIV-1 virus during the production step. During the infection, when the HIV-1 virions successfully fuse their membrane with the plasma membrane of the host cell the BlaM is released to the cell cytoplasm where it cleaves the CCF2 molecules (composed of two fluorophores constituting a FRET pair). This cleavage is typically monitored by measuring a spectral shift of the CCF2 fluorescence emission or a ratio of the fluorescence intensity of the donor and acceptor. In this article the authors propose a more accurate readout based on the analysis of the fluorescence lifetime.
General questions:
In my opinion the article lacks a clear description of the main principle of the BalM method used for the monitoring of the HIV-1 fusion as well as of the experimental approaches. I suggest to include in the introduction or at the very beginning of the results section, a clear scheme of the cell infected by Vpr-BalM containing virus, the release of the BalM upon the viral fusion and the cleavage of CCR2. Also the authors don’t mention the FRET pair (hydroxycoumarin/fluorescein) of the CCR2.
We thank the reviewer for pointing out the lack of information about CCF2-AM in the manuscript. This point has been now addressed (see response to reviewer 1, point 3). We do state now that hydroxycoumarin is the donor and fluorescein the acceptor in the CCF2 biosensor and explain the catalytic reaction in the presence of BlaM enzyme.
In the figure 2 and 3 the authors quantify the number of viral particles in the cells at 90 minutes +2h post infection using a calibration curves based on the MOI inferred from the expression of b-gal in TZM-bl cells 48 hours post infection (with 24 hours of virus/cell contact). This implies that all the viral particles enter the cell within the first hours of infection?
Indeed, we have performed in the past several kinetic studies in which we show that 90-120 min is enough to reach the equilibrium for the HIV fusion reaction [Jones et al. 2017 Cell Reports or Miyauchi et al. 2009 Cell]. We cite now these papers and discuss this circumstance.
We have now toned down our quantitative claims and acknowledge the fact that what we calculate is the relative number of HIV paritcles internalized per cell and not the absolute number of infected particles which would be more related to the MOI.
Are the % of infected cells the same is the viruses are washed out after initial 90 minutes incubation? Could the authors explain better the relationship between the MOI measured at 48 hours post infection and the number of intracellular viral particles determined in their experiments?
Yes, as reported previously if one allows the virus to sediment down during 60 minures, and then one wahes the sample to avoid second or third waves of infection the number of fusogenic cells is very similar as compared to the infected ones.
When two population analysis is used for the FLIM/FRET data, were all the fit parameters free or one of the lifetimes was fixed?
We thank re reviewer for this comment. We have now included the mathematical description of the model utilized for fitting the FLIM images with a double exponential methods L139 – L177
The data analysis should be better described in the text. Part 3.2 and 3.3 lack a clear description of the experimental procedure and repeatability of the experiments. How many independent samples were prepared? How many cells were analyzed in each condition?
This point has now been addressed and specified in each figure legend.
In the part 3.3 the authors determine the number of viruses that successfully enters the T macrophages using a calibration curve that was done with TZM-bl cells? However depositing the same quantity of the virus on the cells the % of infection, and thus the calibration curve is different for the T Cells comparing to TZM-bl cells.
We apologize the reviewer for the confusion. The method presented here indeed does not allow to determine absolute number of internalized viral particles per cell. We determined the linear dependence between amount of inoculated viruses and the cleavage of CCF2-AM, allowing estimation of the relative amount of internalized viral particles per cell independently of the cell line used in the study.
In the data in figure 1, values presented for the ratiometric and fraction donor approach are lower in the first infected sample (MOI 1) than the for the “No envelope” control. This point should be commented.
Indeed, the ratiometric approach here seems to be more noisy as compared to the FLIM (fD). We have tried to discuss this circumstance throughout the text, but the only explanation we could find is that intensity based approaches are noiser because of two things: first they give more background and second they are concentration dependent (as opposed to FLIM (fD)). We discuss this in the introduction and material and methods.
Concerning the data in figure 3, it is not clear why does the donor lifetime decrease when the HIV-1 viruses enter into the T Cells? The fusion should be accompanied by an increase of the average donor lifetime due to the loss of the FRET. This point should be better commented since it compromises the validation of the method.
As all the experiments were acquired utilizing a time-correlated single photon counting (TCSPC) approach we have now simplified the article and only show the model with two exponentials and the calculation of the fraction of interacting donor (CCF2, hydroxycoumarin engaged in FRET with fluorescein).
Moreover, in the case of the T cells (New Figure 4) the diminution of the apparent average lifetime came from an increased background that is not corrected. We understand that the Leica software calculates the dealy of each single photon relative to the laser pulse and builds on-line a histogram that when the S/N is low (around 100-200 photons) could be unreliable. Especially because one cannot substract the background photons. We have now simplified the approach only presenting the fitting approach for both exmples (TZM-bl cells and T Cells).
We have also updated the material and methods so that a better explanation of the mathematical description for both the catalytic reaction and the fluorescence decay can be followed by the average reader.
Minor points:
Material and Method Section: The source of the HXB2 envelope is missing Annotations of the inserts (A,B,C….) in figures 1 and 3 are missing. In the figure 1 I suggest to respect the order of the presentation of the three analysis approaches in the micrographs and in the statistics. i.e. the cell images are organized from the left to the right D/A intensity ratio, Donor lifetime, fraction of the donor, then the graphs should follow the same order from the top to the bottom. For the better understanding of the cell images in the figure 1 I suggest to add the colorcode as it is done in figures 2 and 3.
We thank the reviewer for these comments and now have been updated accordingly.
The manner to define the threshold (green dotty line) for the discrimination of the fusion+ cells is not clear. Authors say that this level corresponds to cells exposed to No Env virions, but the values corresponding to these No Env virions are also present in the figures as a negative control in line 1. Does the threshold represent the max (or min for the f(D)) value measured for the No Env? This point should be better explained in the text.
We apologize the reviewer for the lack of explanation. We have now described in further detail the criteria to define the threshold in the results section of figure 2.
Round 2
Reviewer 1 Report
The authors satisfactory addressed all my questions/comments.
Author Response
N/A
Reviewer 2 Report
In my view the revisions that the authors made based on the reviewers comments have improved the manuscript. However, I still have concerns regarding what one would call validation. This is a paper on a method with a slightly complex readout and relatively minor shifts between the different levels of infection. Therefore I would have expected that the experiment shown in revised figure 2 - which is the exclusive foundation of the approach - has been done more than once (i.e. three independent experiments) in order to demonstrate soundness and robustness of the method. According to the information in the revised version, this was not the case. It is not clear to me why this has not been done to ensure the quality of the work.
The experiment shown in figure 3 has been performed three times, but as far as I understand the figure (it is not entirely clear from the legend) results from all 3 replicates are pooled for the graphs, so that it is not clear whether there is variation between the replicates.
While the authors have toned down the quantitative statements throughout the text to make clear that what was measured are relative values, they have retained what I would consider to be overstatements in the title and abstract. The experiment with primary T-cells was performed a single time with primary cells from only a single donor, and the calibration was taken from a single experiment in a different cell line. Based on this experiment and 10 lines of text it appears to me very misleading to emphasize primary T-cells by mentioning them in the title of the manuscript. Also, the last sentence of the abstract is my opinion an overstatement – experiments shown in figure 3 and 4 demonstrate that the method has been applied to both cell types, but in the present form do not provide any validation. The title and this sentence should be changed to reflect the content of the mansucript.
Minor point: this may seem petty, but what the authors have done to revise the list of references is not exactly what I had in mind when suggesting a more comprehensive coverage of the background and the work of others. Similarly, citing the person who provided plasmids rather than the labs that generated them (I do not have a personal interest) is in my view not the current standard.
The manuscript still contains some typos.
Author Response
Reviewer 2
In my view the revisions that the authors made based on the reviewers comments have improved the manuscript. However, I still have concerns regarding what one would call validation. This is a paper on a method with a slightly complex readout and relatively minor shifts between the different levels of infection. Therefore I would have expected that the experiment shown in revised figure 2 - which is the exclusive foundation of the approach - has been done more than once (i.e. three independent experiments) in order to demonstrate soundness and robustness of the method. According to the information in the revised version, this was not the case. It is not clear to me why this has not been done to ensure the quality of the work.
We thank the reviewer for this comment and have now included the data coming from three different experiments.
The experiment shown in figure 3 has been performed three times, but as far as I understand the figure (it is not entirely clear from the legend) results from all 3 replicates are pooled for the graphs, so that it is not clear whether there is variation between the replicates.
While the authors have toned down the quantitative statements throughout the text to make clear that what was measured are relative values, they have retained what I would consider to be overstatements in the title and abstract. The experiment with primary T-cells was performed a single time with primary cells from only a single donor, and the calibration was taken from a single experiment in a different cell line. Based on this experiment and 10 lines of text it appears to me very misleading to emphasize primary T-cells by mentioning them in the title of the manuscript.
We have stressed that what is measured is the relative number of particles that were fusion positive per cell. This also includes primary T cells. We have now toned down the message both in the title and the abstract stressing that we have performed these analyses in live cells. Importantly, as stated in Zotter et al (now referenced in the paper) the BlaM cleavage reaction does not differ between different cell lines but it does when compared in vitro. Indeed, as discussed in Zotter et al; showed that k(cat)/KM negatively scales with [E] in live cells. In vitro, k(cat)/KM is independent on enzyme concentration (and thus the catalytic reaction linearly scales with [E]). The main difference when comparing the BlaM catalytic reaction in live cells and in solution is the slower diffusion of the substrate CCF2 in the cytosol of cells. The catalytic reaction also turned out to be independent of the metabolic state and the B-Lac mutant employed. The diffusion of these small molecules should not change significantly from cell type to cell type and therefore the calibration performed in TZM-bl for CCF2 cleavage should not be cell-type dependent. We have now discussed this and cited the literature referred here l143 – 148.
Also, the last sentence of the abstract is my opinion an overstatement – experiments shown in figure 3 and 4 demonstrate that the method has been applied to both cell types, but in the present form do not provide any validation. The title and this sentence should be changed to reflect the content of the mansucript.
We have followed the reviewer’s advice and have employed the term “applied” instead of “validated” in the abstract and also changed the title.
Minor point: this may seem petty, but what the authors have done to revise the list of references is not exactly what I had in mind when suggesting a more comprehensive coverage of the background and the work of others. Similarly, citing the person who provided plasmids rather than the labs that generated them (I do not have a personal interest) is in my view not the current standard.
We thank the reviewer for this remark and have endeavoured to cite the source of the plasmids and also the original papers (obtained from the NIH repository). We hope we have found the original articles the reviewer is referring to.
The manuscript still contains some typos.
We thank the reviewer for this comment and have tried to correct those in this final version.
Reviewer 3 Report
The article has been considerably improved and I recommend its publication. However I have just one last minor remark. The authors should verify and correct the mismatch between the references to the figure 2 in the text (page 6, l 207, 211,217…etc) and the real annotations of the panels in the figure 2.
Author Response
Reviewer 3
The article has been considerably improved and I recommend its publication. However I have just one last minor remark. The authors should verify and correct the mismatch between the references to the figure 2 in the text (page 6, l 207, 211,217…etc) and the real annotations of the panels in the figure 2.
We thank the reviewer for this comment and have now corrected the references